# Electron iso-density surfaces provide a thermodynamically consistent representation of atomic and molecular surfaces

Amin Alibakhshi [1,2,3,4] ✉ & Lars V. Schäfer [1] ✉

The surface area of atoms and molecules plays a crucial role in shaping many physiochemical properties of materials. Despite its fundamental importance, precisely defining atomic and molecular surfaces has long been a puzzle. Among the available definitions, a straightforward and elegant approach by Bader describes a molecular surface as an iso-density surface beyond which the electron density drops below a certain cut-off. However, so far neither this theory nor a decisive value for the density cut-off have been amenable to experimental verification due to the limitations of conventional experimental methods. In the present study, we employ a state-of-the-art experimental method based on the recently developed concept of thermodynamically effective (TE) surfaces to tackle this longstanding problem. By studying a set of 104 molecules, a close to perfect agreement between quantum chemical evaluations of iso-density surfaces contoured at a cut-off density of 0.0016 a.u. and experimental results obtained via thermodynamic phase change data is demonstrated, with a mean unsigned percentage deviation of 1.6% and a correlation coefficient of 0.995. Accordingly, we suggest the iso-density surface contoured at an electron density value of 0.0016 a.u. as a representation of the surface of atoms and molecules.

The molecular surface plays a fundamental role in many physiochemical properties, implying the importance of an accurate characterization of the surfaces of atoms and molecules in a wide range of scientific fields. In a computational context, the molecular surface plays a pivotal role in continuum solvation models[1-3], which are widely used in theoretical and computational chemistry to implicitly include solvent effects, in the study of non-covalent interactions[4-8], van der Waals (vdW) materials[9-12], water-octanol partitioning[13], dispersion interactions in DFT computations[14], thermodynamics of phase changes[15], molecular recognition and reactivity[16-18], crystallization[19],

cavity prediction for molecular docking[20,21], and many other applications. Investigating atomic and molecular surfaces and their influence on material properties is a topic with a long history. In his pioneering work, Meyer reported already in 1870 a periodic trend in density attributed to the variations in atomic volumes[22]. This seminal work was the cornerstone of conceiving the vdW surfaces and their quantification via crystallography-based experiments by Bragg[23], Pauling[24], Kitaigorodskii[25], and Bondi[26,27], followed by other experimental methods for estimating vdW surfaces, e.g., via studying the distances of different elements from a probe atom[28], ionization potential[29],

[1]Center for Theoretical Chemistry, Ruhr University Bochum, 44780 Bochum, Germany. [2]Lehrstuhl für Theoretische Chemie II, Ruhr University Bochum, 44780 Bochum, Germany. [3]Research Center Chemical Sciences and Sustainability, Research Alliance Ruhr, 44780 Bochum, Germany. [4]Research Center Trustworthy Data Science and Security, Technical University Dortmund, 44227 Dortmund, Germany. ✉e-mail: amin.alibakhshi@ruhr-uni-bochum.de; lars.schaefer@ruhr-uni-bochum.de

thermodynamics of bond breakage[30], and models employing solid molar volumes[31]. Although being insightful and extensively used in many different applications due to the ability to provide a rough estimation of atomic and molecular surfaces, the heuristic definition of the vdW surfaces typically introduces ambiguities. This is reflected in the existence of multiple variants of vdW surfaces, such as solvent-excluded or solvent-accessible ones, as well as multiple parameterizations of atomic radii for them, resulting in a broad range of estimations for molecular surfaces[15]. The diversity of available parameterizations of atomic vdW radii stems from uncertainties in experimental approaches, which commonly estimate vdW radii from interatomic distances. For crystallography-based experiments, a systematic underestimation of radii is commonly expected due to the stronger interaction between atoms in solids[32–34]. For methods that study atoms in heteronuclear molecules, the estimated radii and resulting surfaces can be affected by bond polarity or anisotropy[35]. Alongside the uncertainties linked to vdW radii parameterization, the construction of molecular vdW surfaces from those radii, due to the assumption of a perfectly spherical shape for the atoms, can also significantly contribute to deviations from the true molecular surfaces. Obviously, for noble gases, the perfect sphere assumption holds and therefore, the vdW surfaces can be a precise representation. Nevertheless, for real molecules, anisotropic distribution of the electron density can result in significant deviations from this assumption. Most importantly, the available parameterizations of vdW radii do not take the dependence of atomic radii on the atomic partial charge into account, which can vary due to the chemical environment. The above-mentioned limitations in characterizing molecular surfaces can be overcome by an elegant theory initially conceptualized by Bader and co-workers. According to this approach, the surface of atoms and molecules is defined as an iso-density surface beyond which the electron density drops below a certain threshold[36]. Therefore, there is no requirement to parameterize the radii of individual atoms or consider atoms as perfect spheres. Instead, this definition of surfaces only requires electron density data, which can be obtained from quantum chemical computations. Despite being robust and elegant, the validation of this method and the appropriate cut-off value of the density has not been amenable to extensive experimental verification so far. As a rough estimate for the cut-off density, Bader and co-workers suggested a value of 0.002 a.u.[36], and Boyd suggested a value of 0.001 a.u.[37], each resulting in significantly different iso-density surfaces. Rahm et al. considered the cut-off density suggested by Boyd for approximating vdW radii of the first 96 elements of the periodic table[38]. One main reason that restricted a more precise quantification of the cut-off density are limitations of experimental methods, as discussed above. Apart from the challenges with experimental methods, constructing the surface of molecules from those radii in a way to properly account for anisotropy in atomic electron densities is not a trivial task. The recent conception of thermodynamically effective (TE) surfaces[15] now enables to re-examine and benchmark the iso-density definition of molecular surfaces. By suggesting a thermodynamically consistent definition for molecular surfaces, TE surfaces allow an experimental evaluation of molecular surface areas from thermodynamic phase-change data. Importantly, unlike other commonly applied methods that are limited to the evaluation of interatomic distances, TE surfaces provide a direct estimate of the total surface area of a molecule, which can then be compared with the iso-density surfaces. Therefore, we select this method in the present study to assess the Bader iso-density theory and to quantify the appropriate value of the density cut-off required in this theory.

## Results and discussion

The first aim was to identify the DFT method that yields the most accurate electron densities, and to employ it for computations with a larger basis set. Therefore, the iso-density surfaces computed using

**Table 1 | Agreement of DFT iso-density surfaces with CCSD(T) results calculated for all studied cut-off densities using the def2-TZVPD basis set. The mean unsigned percentage error (MUPE) and Pearson correlation coefficient (R) are given**

| Method | MUPE (%) | R |
|---|---|---|
| PBE | 0.24 | 0.9998 |
| B3LYP | 0.36 | 0.9998 |
| DSD-PBEP86 | 0.08 | 0.9999 |

different DFT methods were compared with those from CCSD(T) computations for the def2-TZVPD basis set. By comparing all DFT iso-density surfaces for all studied cut-off densities (from 0.0008 to 0.0025 a.u., with 0.0001 a.u. intervals) with the respective CCSD(T) iso-density surfaces, we found the best agreement for the double-hybrid DSD-PBEP86 functional showing only 0.08% average absolute deviation (Table 1).

Accordingly, we selected this double-hybrid functional for computations of iso-density surfaces with the larger quadruple-zeta basis set and refer to these DSD-PBEP86/def2-QZVPD calculations in the rest of this study for investigating iso-density surfaces. As a further verification, we also computed iso-density surfaces at the CCSD(T)/def2-QZVPD level of theory, which due to the computational demands could be completed for the lowest energy conformers of a subset of 27 compounds. For them, a comparison of iso-density surfaces between DSD-PBEP86 and CCSD(T) methods, both with the def2-QZVPD basis set, yielded a MUPE of 0.16% and a correlation coefficient of 0.99999, underlining the robustness of the double-hybrid DFT results.

Comparing the theoretically calculated iso-density surfaces (at DSD-PBEP86/def2-QZVPD level of theory) with experimentally derived TE surfaces as reference values for the 104 studied molecules, the best agreement was observed for the cut-off density of 0.0016 a.u., which yielded MUPE and Pearson correlation coefficient of 1.59% and 0.995, respectively. Details of the computed surfaces are provided in the supplementary material.

To further investigate the robustness of this suggested value of the cut-off density, we also evaluated the optimum cut-off densities for 10 datasets, each containing 50 randomly selected compounds from the benchmark set. For all cases, we found the value of 0.0016 a.u. to yield the best agreement between TE and iso-density surfaces.

Furthermore, for a larger dataset containing 184 additional compounds with higher uncertainty in the experimental phase-change data, the best agreement between TE and iso-density surfaces was observed for electron density cut-off of 0.0015 a.u. with a MUPE of 6.5%. This result was close to the MUPE of 6.6% obtained using a cut-off density of 0.0016 a.u. (Pearson correlation coefficient is 0.94 in both cases). Despite the larger experimental uncertainties for this dataset, the agreement between the found optimum cut-off densities for the benchmark set and the test set is very comparable. We thus interpret these results as an additional support of the robustness of the suggested cut-off density of 0.0016 a.u.

Figure 1 shows the agreement of the molecular iso-density surfaces with the TE surfaces as a function of the density cut-off. Density cut-offs in the range 0.0015 to 0.0019 a.u. provide the most accurate results, with an optimum at 0.0016 a.u. The cut-off densities of 0.002 a.u. proposed by Bader and co-workers[36] and 0.001 a.u. by Boyd[37] yield MUPEs of 3.17% and 6.98%, respectively, showing the impact of the cut-off density on the computed surfaces and their agreement with the experimental values.

Figure 2 shows correlation plots of the molecular surfaces obtained via the two different approaches, experimentally derived (TE) and iso-density surfaces from DSD-PBEP86/def2-QZVPD computations. Different iso-density surfaces are shown, with cut-off densities of 0.0016 a.u. suggested by us (upper panel), 0.002 a.u. as suggested by

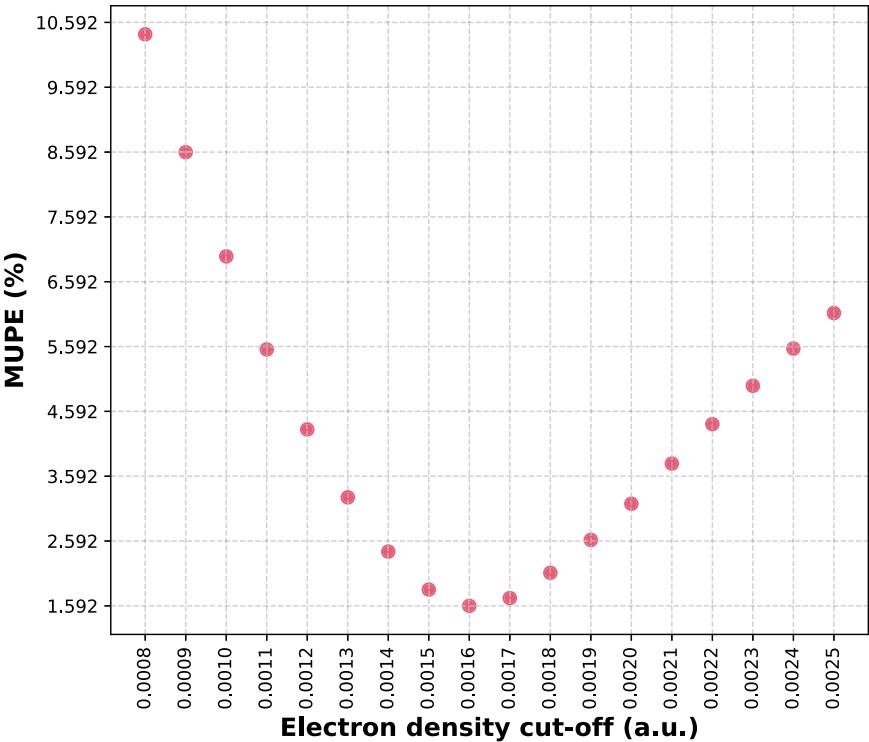

**Fig. 1 | Agreement of iso-density surfaces from quantum chemical calculations with experimental results.** The mean unsigned percentage error (MUPE) is plotted as a function of the electron iso-density cut-off. Source data are provided as a Source Data file.

Bader (middle panel), and 0.001 a.u. as suggested by Boyd (lower panel). The cut-off density of 0.0016 a.u. provides the best agreement, while the two other cut-off densities applied yield slightly larger deviations. The close to perfect correlation ($R = 0.995$) for all density cut-offs applied underlines the close correspondence between the TE and iso-density surfaces. Considering that the employed experimental TE approach for quantifying molecular surfaces is based on thermodynamic phase-change data while the theoretical results are obtained from the computed electron densities and thus in a completely different way, this remarkable agreement is interpreted as a strong mutual validation of both approaches.

Further analysis of the results demonstrates the importance of conformer sampling and Boltzmann averaging. When considering only the single lowest energy conformer of each molecule, the best results were obtained again for the cut-off density of 0.0016 a.u., but the agreement to the TE surfaces (MUPE of 1.87%) was lower than those obtained via conformer sampling (MUPE of 1.59%). The importance of conformer sampling becomes more evident when studying larger and more flexible molecules, which have a greater number of low-energy conformers. Accordingly, in the studied set of molecules, the three most pronounced improvements by conformer sampling were observed for 2-methyl-octane, heptane, and 1-hexanethiol, where the agreement with TE surfaces for single lowest energy conformers in terms of MUPE were reduced via conformer sampling (Boltzmann averaging) from of 2.30, 4.21, and 2.9% to 0.44, 2.12, and 0.19%, respectively. An illustration of the importance of Boltzmann averaging and the variation of iso-density surfaces for two low-energy conformers is depicted in Fig. 3 for 1-pentanethiol as an example.

One of the important applications of the proposed method is the evaluation of atomic radii of elements. In Table 2, we present the atomic radii of a number of elements and their comparison with experimental estimations. Details of the computations, including employed phase-change data are provided in supplementary material.

Table 2 shows that for noble gases (for which the uncertainty in experimental estimations are significantly lower, as discussed above),

there is an excellent agreement between the atomic radii estimated via TE surfaces, iso-density surfaces with cut-off density of 0.0016 a.u., and reference values from Ref. 39. Also, for N and F elements, a close agreement between the radii determined via iso-density surfaces and the experimental estimation is observable. For the radii estimated with iso-density cut-off of 0.001 a.u., the results closely match the values reported by Rahm et al. calculated for the same cut-off density at PBE0 level of theory[38]. However, the computed radii with cut-off density of 0.001 a.u. are significantly larger than those predicted by the other studied methods, highlighting the importance of the applied density cut-off. Noteworthy, the estimations of atomic radii for N, O, and F elements via iso-density surfaces are obtained from the diatomic molecule. In another study[40], we recently proposed a rigorous approach to estimate radii of open-shell atoms in their isolated states on the basis of the Tkatchenko-Scheffler method[14] relating radii of atoms in molecules to the radii of free atoms via:

$$R_A = R_{A,free} \left( \frac{V_A^{eff}}{V_A^{free}} \right)^{1/3}, \qquad (1)$$

$$\frac{V_A^{eff}}{V_A^{free}} = \frac{\int r^3 w_A(\mathbf{r}) n(\mathbf{r}) \, d^3\mathbf{r}}{\int r^3 n_A^{free}(\mathbf{r}) \, d^3\mathbf{r}}, \qquad (2)$$

where $w_A$ is the atomic weight for partitioning the molecular space into atomic sub-spaces, and $R_{A,free}$ and $n_A^{free}$ are the radius and the electron density of atom $A$ in the free (isolated) state.

Using iso-density surfaces with a cut-off density of 0.0016 a.u., we optimized the radii of free atoms in a way to obtain the best match between solvent-excluded surfaces (SES) constructed from the radii of atoms in molecules based on Eq. (1) and iso-density surfaces for a set of 1235 molecules[40]. The molecular surfaces predicted on the basis of these two different approaches closely match, with MUPE of 0.75% and Pearson correlation coefficient of 0.9996. Also, the optimized radii of N and O atoms, found to be 1.65 Å and 1.49 Å, respectively, perfectly

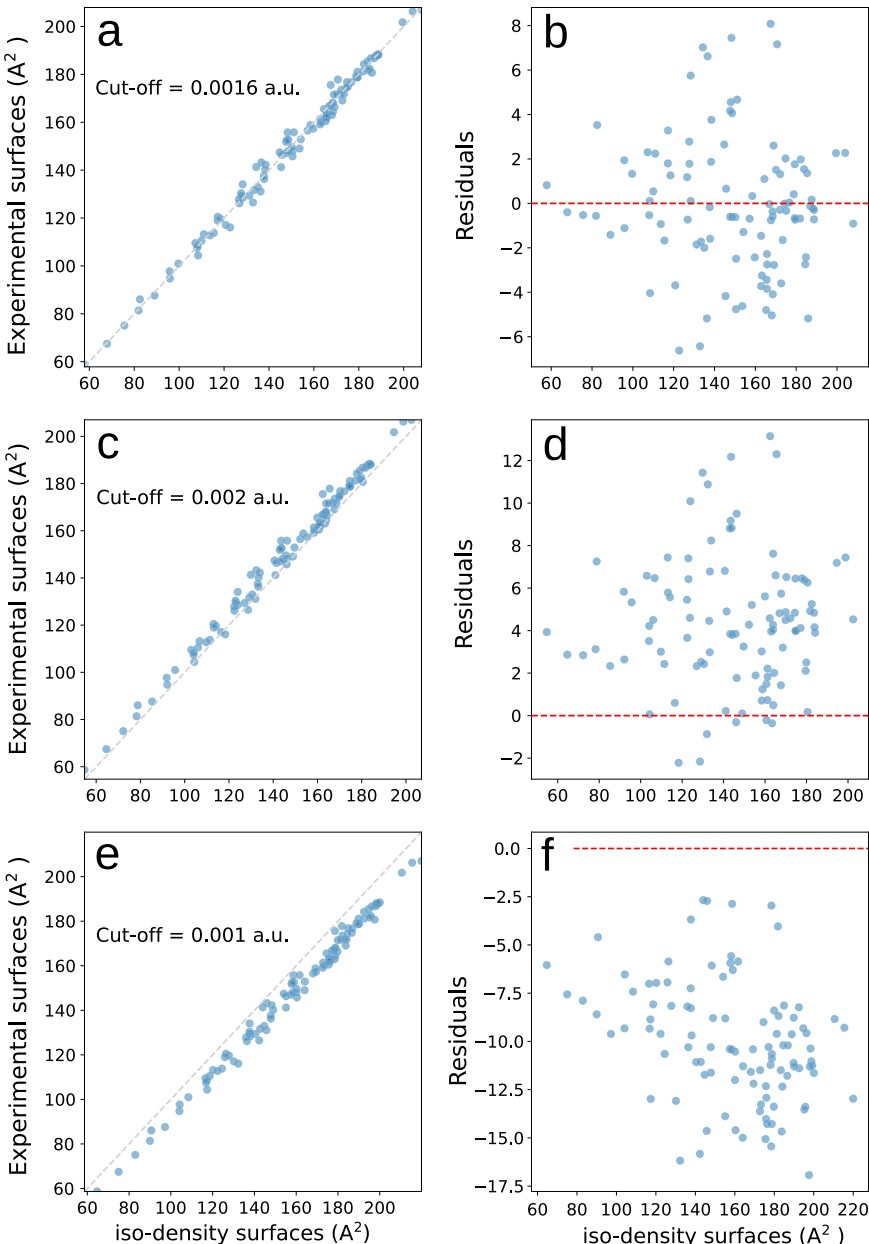

**Fig. 2 | Comparison of iso-density surfaces (in Å²) from quantum chemical calculations with experimental surfaces. a**, **b** Cut-off density 0.0016 a.u. **c**, **d** 0.002 a.u. **e**, **f** 0.001 a.u. The residuals are plotted at the right. Source data are provided as a Source Data file.

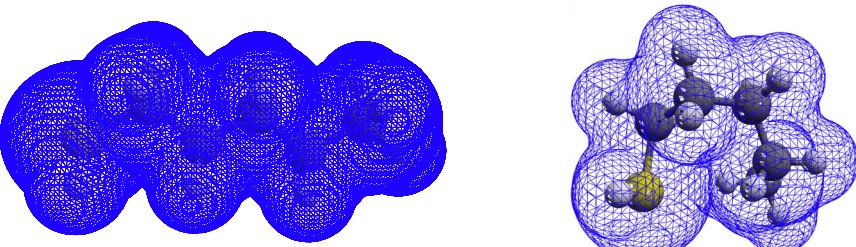

**Fig. 3 | Iso-density surfaces of two low-energy conformers of 1-pentanethiol contoured at cut-off density of 0.0016 a.u.** The surface areas of 159.8 Å² (left) and 149.2 Å² (right) are significantly different from the experimentally determined surface (156.5 Å²). The Boltzmann averaged iso-density surface for multiple conformers is 157.2 Å², in agreement with the experimental value.

**Table 2 | Comparison of different estimations of atomic radii of selected elements (in Å). Experimental estimations of atomic radii for noble gas atoms are taken from Ref. [39]. and for other elements from Ref. [28]**

| | TE | Exp. | iso-density (cut-off 0.0016 a.u.) | iso-density (cut-off 0.001 a.u.) | iso-density (cut-off 0.002 a.u.) |
|---|---|---|---|---|---|
| Ar | 1.92 | 1.94 | 1.87 | 1.98 | 1.81 |
| Kr | 2.04 | 2.07 | 2.01 | 2.13 | 1.96 |
| Xe | 2.23 | 2.28 | 2.20 | 2.33 | 2.14 |
| N | 1.76 | 1.66 | 1.69 | 1.79 | 1.64 |
| O | 1.60 | 1.50 | 1.61 | 1.71 | 1.56 |
| F | 1.50 | 1.46 | 1.52 | 1.61 | 1.48 |

agree with the experimental estimations (1.66 Å and 1.50 Å, Table 2). The consistency of the two alternative computational methods for evaluating molecular surfaces and the agreement of the optimized radii with the experimental estimations strongly reinforce the robustness of the proposed approaches.

## Methods

The agreement between TE and iso-density surfaces was benchmarked against the DIPPR database[41]. We only considered compounds in that database for which the reference thermodynamic phase-change data of both vaporization enthalpy and surface tension were measured experimentally and with an uncertainty below 3%, resulting in an initial dataset of 171 compounds. Considering that in the evaluation of TE surfaces, molecules are assumed to follow the ideal gas law in their vapor state[15,42], we further refined this dataset by excluding molecules that are susceptible to form halogen or hydrogen bonds in the gas phase and thus might significantly deviate from an ideal gas due to cluster formation. Empirical modification of the in-vacuo calculated enthalpies to correct for clustering in the vapor state was demonstrated in a previous study to be necessary for high accuracy estimation of combustion enthalpy[43]. Excluding these compounds yielded a final dataset of 104 compounds, which are listed in Table S2 in supplementary information. In addition to this refined dataset, which serves as the benchmark set in this study, we also evaluated TE and iso-density surfaces for compounds with an uncertainty in the experimental phase-change data between 3% and 5%, yielding an additional dataset of 184 compounds. The details of the computed TE and iso-density surfaces for this larger set of compounds is reported in Table S3 in supplementary information.

For the studied molecules, experimental values of molecular surface areas ($a_s$) were obtained by employing the following equation, which describes their relationship to experimental phase-change data of vaporization enthalpy ($\Delta H_{vap}$), surface tension ($\gamma$), and critical temperature ($T_c$)[15].

$$\Delta H_{vap} = \frac{a_s}{2}\left(2\gamma - T\frac{d\gamma}{dT}\right) - \frac{RT}{2}\ln\left(\frac{T}{T_c}\right) \quad (3)$$

We considered phase-change data at 25 temperature points linearly distributed between the melting point and the critical temperature of each compound and found the $a_s$ values by fitting. The temperature dependence of the surface tension and its analytical derivatives at different temperatures were estimated using the Guggenheim–Katayama relationship[44]:

$$\gamma(T) = \gamma°\left(1 - \frac{T}{T_c}\right)^{11/9} \quad (4)$$

The thus obtained molecular surfaces are reported in Table S2.

For the theoretical evaluation of molecular surfaces via the iso-density criterion, we computed the electron densities by ground-state electronic structure computations. Considering that in many molecules, typically multiple low-energy conformers contribute to the experimental results, we generated up to 25 low-energy conformers for each molecule using the CREST method[45]. This procedure yielded a total number of 1071 conformers for the 104 studied molecules.

For each conformer, the electron density distribution was acquired via electronic structure computations. For that, the coupled cluster CCSD(T) level of theory is expected to provide the highest accuracy, as this method is considered as the gold standard in quantum chemistry. CCSD(T) calculations generally require large basis sets, and computational costs can be very high. As an alternative, DFT methods can be employed, as they typically provide high accuracy electron densities that may not significantly deviate from the coupled cluster ones despite being obtained at a much lower computational cost. Accordingly, Rahm et al. considered PBE0 computations for evaluating vdW radii based on iso-density surfaces at cut-off density of 0.001 a.u.[38]. In the present study, to acquire accurate electron densities close to complete basis set limit, we first performed electronic structure computations at PBE, B3LYP, DSD-PBEP86[46], and CCSD(T) levels of theory with the def2-TZVPD basis set. For the DSD-PBEP86 double-hybrid DFT method, which yielded the best agreement with the CCSD(T) computations as demonstrated in the results section below, we recomputed electron densities with the larger def2-QZVPD quadruple-zeta basis set. The final molecular surfaces were considered as the Boltzmann average of all surfaces computed with the quadruple-zeta basis set for the different conformers of each molecule.

All electronic structure computations were carried out with Orca 5.0.3[47]. Using the generated wavefunctions for each conformer, total iso-density surfaces were computed for different cut-off densities ranging from 0.0008 to 0.0025 a.u. with 0.0001 a.u. intervals based on an improved marching tetrahedra algorithm[48] developed by Lu and Chen[17] implemented in the Multiwfn software[49].

In addition to the total surface areas of the studied molecules, we also investigated the accuracy of predicted atomic radii of certain elements using TE and iso-density surfaces. For noble gases, where the surfaces are uniquely defined via vdW radii and the perfect sphere assumption holds, estimation of atomic radii from the evaluated TE or iso-density surface is straightforward. For other elements, to avoid undesired effects due to bond polarity, anisotropy, or the radii dependence on atomic partial charges, we only considered homonuclear diatomic molecules $N_2$, $O_2$, and $F_2$ for which phase-change data at different temperatures are available in the NIST database[50]. For these molecules, using reference bond lengths taken from the NIST database and the evaluation of the total surface area of the molecule, atomic radii can be estimated using simple geometrical considerations (see supplementary information for details). We did not consider He, Ne, and $H_2$ in this analysis, because due to their very low boiling points (below $-246\,°C$), high uncertainties in thermodynamic phase-change data (and thus in the resulting TE surfaces) are expected.

The agreement between the theoretically estimated molecular surfaces in comparison to the experimental values is reported in terms of mean unsigned percentage error (MUPE) and Pearson correlation coefficient (R),

$$\text{MUPE} = \frac{1}{N}\sum\left(\left|\frac{y_{i,1} - y_{i,2}}{y_{i,1}}\right|\right) \times 100 \quad (5)$$

$$R = \frac{\sum(y_{i,1} - \bar{y_1})(y_{i,2} - \bar{y_2})}{\sqrt{\sum(y_{i,1} - \bar{y_1})^2 \sum(y_{i,2} - \bar{y_2})^2}} \quad (6)$$

where $y_{i,1}$ and $y_{i,2}$ are TE and iso-density surfaces of molecule i, respectively, and $\bar{y_1}$ and $\bar{y_2}$ are the corresponding sample means. For

the homonuclear diatomic molecules, details of the studied phase-change data are provided in Table S1, together with Orca and Multiwfn scripts to calculate iso-density surfaces.

In summary, we used quantum chemistry calculations to compute iso-density surfaces for 104 molecules and compared them with accurate experimental estimations based on TE surfaces. Our results show a close agreement between iso-density surfaces for a cut-off density of 0.0016 a.u. and TE surfaces derived from experimental phase-change data with MUPE of 1.59% and correlation coefficient of 0.995. The suggested cut-off density accurately reproduces the bonding radii of noble gases. Based on these results, we conclude that an iso-density surface around atoms and molecules contoured at an electron density value of 0.0016 a.u. reliably represents atomic and molecular surfaces.

## Reporting summary

Further information on research design is available in the Nature Portfolio Reporting Summary linked to this article.

## Data availability

Source data are provided with this paper.

## Code availability

Sample code to carry out the computations is provided in supplementary information and as Supplementary Software.

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

## Acknowledgements

This work was supported by the Deutsche Forschungsgemeinschaft (DFG) under Germany's Excellence Strategy - EXC 2033 – 390677874 - RESOLV. We gratefully acknowledge the funding of this project by computing time provided by the Paderborn Center for Parallel Computing (PC2).

## Author contributions

AA conceived and designed the study, carried out the computations, contributed to the analysis and interpretation of the data, and wrote the first draft of the manuscript, which was then edited by both authors jointly. LVS contributed to the analysis and interpretation of the data and writing the manuscript.

## Funding

## Competing interests

The authors declare no competing interests.
