## [Peer Review File · Nature Communications]

Electron Iso-Density Surfaces Provide a Thermodynamically Consistent Representation of Atomic and Molecular SurfacesREVIEWER COMMENTS

Reviewer #1 (Remarks to the Author):

Alibakhshi and Schäfer derive an elegant way to define the Bader iso-density surface that is compatible with experimental phase change data. This is an interesting approach and the results are well supported by the data. This referee is positive to publication if the comments below are addressed.

The choice of test compounds is limited to just hydrocarbons, which are rather different from compounds mentioned the range of applications in the introduction (ref 1-17). Authors should therefore extend their dataset with many more compounds. The Dippr database contains vaporization enthalpies for close to 2000 compounds, and surface tensions for most of these compounds as well. If non-ideality in the gas-phase needs to be neglected authors can just use temperatures well above the freezing point.

From a data science perspective, authors have a training set but no independent test set (except for F₂, N₂, O₂). This is simply not acceptable today. Taken together with the chemical argument above, authors should extend their data set with different compounds and split the data into independent training and testing compounds.

Authors write that "... this definition of surfaces only requires electron density data, which can be obtained from quantum chemical computations.", but there is a practical problem in the "only". The range of molecules for which a sufficiently accurate density can be readily computed is rather limited. Could there be alternative methods for calculating the electron density, more accurate than just based on radii, but less costly than quantum chemistry?

Indeed, authors compare radii compatible with the TE method and DFT to the well-known Bondi radii, and describe the results (Table 3) as "excellent agreement" although the Bondi radius for N is 10% lower than the other two. This despite the fact that they write in the introduction that "multiple parameterizations of atomic radii ... resulting in a broad range of estimations for molecular surfaces" which to this referee suggests that 10% difference is not negligible. Also, how about radii of atoms in real compounds? Can authors define a single radius for carbon and hydrogen that reproduces the DFT surface? If not, what is the point of going back to radii?

Details:

It is strange that authors use a triple zeta benchmark for validating their choice of DFT methods to CCSD(T) and then ad-hoc change the basis set to a quadruple zeta basis set. Authors should add a comparison to CCSD(T) for this combination to Table 1 and evaluate the change of basis set in the same manner as the choice of DFT method.

Table 2 can be moved to the supporting information.

Figure 2 would be better shown as residual plots to allow readers to see whether the differences between model and experiment are systematic and whether there are outliers.

The caption of Table 3 fails to mention N₂, O₂, F₂.

Table S1: the values of DH_{vap} seem to be a factor 1000 too high.

Reviewer #2 (Remarks to the Author):

This paper was a pleasure to read, and the results obtained by the authors pointing to the 0.0016 au iso-density surface for defining atomic or molecular surfaces as a reliable one for obtaining surfaces areas is important. I find no flaws in the data analysis, interpretation of the results and the stating of the conclusions.

My main comment is the lack of mention of earlier work by Politzer et al relating to atomic and molecular surfaces, areas and volumes. See, for example: J. Org. Chem. 58, 7070 (1993); J. Mol. Struct. (Theochem) 307, 55 (1994); J. Mol. Struct. (Theochem) 425, 107 (1998). Another paper focusing on the surface approach is by Bulat et al, J. Mol. Model. 16, 1679 (2010). The authors may also want to look at WIREs Comput. Mol. Sci. 1, 153 (2011).

Reviewer #3 (Remarks to the Author):

It is interesting to see the ideas of Bader connecting the electron density distribution to Van der Waals radii (and the subsequent work by Boyd and Rahm) confronted to experimental data, as done in this paper. The agreement between the experimentally-derived molecular surfaces and the calculated 0.0016 e⁻ isodensity surfaces seems excellent.

I have two objections, though, related with Table 3:

1) The title speaks of "noble gases", while we find in the table N, O and F, besides Ar, Kr and Xe. Do those entries refer to the corresponding diatomic gases? The "noble gases" in the title is at least inaccurate.

2) I understand that the tendency to use the old Bondi radii still persists, but the values of the first column of Table 3 seem better when compared to the most recent radii set reported by Alvarez and Vogt, which moreover include many more elements than Bondi's. Those radii, with the TE values in parentheses) are 1.94 (1.92) , 2.07 (2.04) and 2.28 (2.23) Å for Ar, Kr and Xe, respectively (Inorg. Chem. 2014, 53, 9260), and 1.66 (1.76), 1.50 (1.60) and 1.46 (1.50) Å for N, O and F (ref. 24 of the manuscript).

Dear Editor, dear Reviewers:

We thank you for the time and efforts spent for evaluating our manuscript and for the very positive and constructive feedback, which helped us to further improve the quality of our manuscript. We followed all suggestions made by the reviewers. In particular, as is also detailed in the point-by-point replies below, we significantly expanded the set of molecules tested (by more than a factor of 2) and we included an independent test set. Furthermore, we added a more extensive discussion on the similarities and differences with traditional atomic radii, as also suggested by Reviewer #3.

We hope that with these additions and revisions, our manuscript can be accepted for publication in *Nature Communications*.

Kind regards (on behalf of both authors),

Lars Schäfer

Reviewer #1

Comment: Alibakhshi and Schäfer derive an elegant way to define the Bader iso-density surface that is compatible with experimental phase change data. This is an interesting approach and the results are well supported by the data. This referee is positive to publication if the comments below are addressed.

Response: We thank the reviewer for the positive evaluation of our work and for the constructive feedback. We provide point-by-point replies to all points raised below.

Comment: The choice of test compounds is limited to just hydrocarbons, which are rather different from compounds mentioned the range of applications in the introduction (ref 1-17). Authors should therefore extend their dataset with many more compounds. The Dippr database contains vaporization enthalpies for close to 2000 compounds, and surface tensions for most of these compounds as well.

Response: DIPPR indeed contains roughly 2000 compounds. However, for the majority of the molecules, at least for vaporization enthalpy and surface tension, high uncertainties in the experimental data are reported. As we now explicitly explain in the revised version of our manuscript, to obtain the most reliable estimation of the cut-off density only compounds for which the reference data had been measured experimentally with uncertainties below 3% were considered. Only 177 molecules satisfied this condition.

Nevertheless, we followed the above suggestion by the reviewer, and in the revised version we now also report and discuss the results obtained for 184 additional compounds, for which the uncertainty of phase change thermodynamic data was between 3 and 5%. Furthermore, as is now also discussed in the revised version of our manuscript, we additionally validated our results by studying a much larger dataset containing 1235 molecules in a parallel study (chemRxiv DOI:

<https://doi.org/10.26434/chemrxiv-2024-5qz9b> , a copy of that work was also uploaded together with this resubmission as suppl. information for reviewers) in which we focused on the different topic of quantifying radii of atoms in molecules and their dependence on atomic partial charges. In that work, a close to perfect agreement between iso-density surfaces based on our proposed cut-off density and the surfaces constructed via radii of atoms in molecules evaluated based on a widely accepted Tkatchenko-Scheffler method was found. The fact that none of the two quite different computational methods required phase change thermodynamic data allowed us to study a more diverse set of molecules, thus also addressing the point raised by the reviewer and the agreement between these two methods also further supports the robustness of the approaches. We discuss this in the last part of the “Results and Discussion” section of the revised manuscript.

Comment: If non-ideality in the gas-phase needs to be neglected authors can just use temperatures well above the freezing point.

Response: An important issue is the non-ideality due to clustering of compounds in the vapor state which might be significant for compounds forming halogen or hydrogen bonds even at high temperatures. We demonstrated the importance of correcting such effects in a previous study (Ref. 41 in the revised manuscript). We further elaborate on it in the revised methods section.

Comment: From a data science perspective, authors have a training set but no independent test set (except for F2, N2, O2). This is simply not acceptable today. Taken together with the chemical argument above, authors should extend their data set with different compounds and split the data into independent training and testing compounds.

Response: We fully agree with this point. Accordingly, in addition to studying the radii of Ar, Kr, Xe, N, O, and F elements (which were in all cases determined via data not used in the benchmark set), we also added cross-validation of the cut-off density in the revised version of our manuscript. Furthermore, we refer to our parallel study that considers a comparison of iso-density surfaces with a totally independent set (see above), i.e., the molecular surfaces based on the Tkatchenko-Scheffler approach for 1235 molecules.

Comment: Authors write that “... this definition of surfaces only requires electron density data, which can be obtained from quantum chemical computations.”, but there is a practical problem in the “only”. The range of molecules for which a sufficiently accurate density can be readily computed is rather limited. Could there be alternative methods for calculating the electron density, more accurate than just based on radii, but less costly than quantum chemistry?

Response: Indeed, quantum chemistry calculations might be quite challenging for larger and/or more flexible molecules. Therefore, we changed the term “only” to “solely”. We do not intend to convey the (fallacious) impression that quantum chemical calculations are “easily done in all cases” – rather, we wanted to say that it is in principle straightforward from a conceptual viewpoint (but not necessarily always from the practical viewpoint). We already discussed the other methods

based on the Tkatchenko-Scheffler approach. Another interesting observation in our other study is an almost perfect linear dependence between radii of atoms in molecules and their partial charges obtained based on either iterative Hirshfeld or MBIS partitioning. This might allow a fast and straightforward estimation of atomic radii directly from partial charges which can be determined using several charge prediction schemes (e.g. CENT method) and without requirement of quantum chemical computations.

Comment: Indeed, authors compare radii compatible with the TE method and DFT to the well-known Bondi radii, and describe the results (Table 3) as “excellent agreement” although the Bondi radius for N is 10% lower than the other two. This despite the fact that they write in the introduction that “multiple parameterizations of atomic radii ... resulting in a broad range of estimations for molecular surfaces” which to this referee suggests that 10% difference is not negligible. Also, how about radii of atoms in real compounds? Can authors define a single radius for carbon and hydrogen that reproduces the DFT surface? If not, what is the point of going back to radii?

Response: This is indeed a good point. Following the comment of Reviewer 3, we noticed that with reference to the more recent literature, a better agreement between experimental and reference values exists, which applies also to nitrogen (this is now included in revised Table 2 (previously Table 3)). Furthermore, we refer to our parallel study as a rigorous method for quantifying atomic radii of other elements, including C and H.

Details:

Comment: It is strange that authors use a triple zeta benchmark for validating their choice of DFT methods to CCSD(T) and then ad-hoc change the basis set to a quadruple zeta basis set. Authors should add a comparison to CCSD(T) for this combination to Table 1 and evaluate the change of basis set in the same manner as the choice of DFT method.

Response: We added further verification of suitability of this extrapolation in the first part of the results section and by providing a comparison to CCSD(T)/def2-QZVPD for a subset of the studied molecules.

Comment: Table 2 can be moved to the supporting information.

Response: We moved that table to the supplementary material.

Comment: Figure 2 would be better shown as residual plots to allow readers to see whether the differences between model and experiment are systematic and whether there are outliers.

Response: We added residual plots to the figure 2.

Comment: The caption of Table 3 fails to mention N₂, O₂, F₂.

Response: Thank you for noticing this issue. We modified it in our revision.

Comment: Table S1: the values of DH_{vap} seem to be a factor 1000 too high.

Response: Thank you for pointing that out. The unit should be J, as you noticed, which we corrected in the revision.

Reviewer #2 (Remarks to the Author):

Comment: My main comment is the lack of mention of earlier work by Politzer et al relating to atomic and molecular surfaces, areas and volumes. See, for example: J. Org. Chem. 58, 7070 (1993); J. Mol. Struct. (Theochem) 307, 55 (1994); J. Mol. Struct. (Theochem) 425, 107 (1998). Another paper focusing on the surface approach is by Bulat et al, J. Mol. Model. 16, 1679 (2010). The authors may also want to look at WIREs Comput. Mol. Sci. 1, 153 (2011).

Response: Thank you for directing our attention to those very interesting works. We now refer to them in the introduction.

Reviewer #3 (Remarks to the Author):

It is interesting to see the ideas of Bader connecting the electron density distribution to Van der Waals radii (and the subsequent work by Boyd and Rahm) confronted to experimental data, as done in this paper. The agreement between the experimentally-derived molecular surfaces and the calculated 0.0016 e⁻ isodensity surfaces seems excellent.

I have two objections, though, related with Table 3:

Comment: 1) The title speaks of “noble gases”, while we find in the table N, O and F, besides Ar, Kr and Xe. Do those entries refer to the corresponding diatomic gases? The “noble gases” in the title is at least inaccurate.

Response: Thank you for noticing this issue. We modified it in our revision.

Comment: 2) I understand that the tendency to use the old Bondi radii still persists, but the values of the first column of Table 3 seem better when compared to the most recent radii set reported by Alvarez and Vogt, which moreover include many more elements than Bondi's. Those radii, with the TE values in parentheses) are 1.94 (1.92) , 2.07 (2.04) and 2.28 (2.23) Å for Ar, Kr and Xe, respectively (Inorg. Chem. 2014, 53, 9260), and 1.66 (1.76), 1.50 (1.60) and 1.46 (1.50) Å for N, O and F (ref. 24 of the manuscript).

Response: Thank you for this very useful suggestion. We corrected the reference atomic radii to these more recent values.

REVIEWERS' COMMENTS

Reviewer #1 (Remarks to the Author):

Authors have responded adequately to my previous comments.

Reviewer #2 (Remarks to the Author):

With the thoughtful modifications of the manuscript made by the authors, I find the paper suitable for publication.

Reviewer #3 (Remarks to the Author):

The authors have addressed the issues raised in my previous report, and I recommend publication of the revised version.